# DYRK1A interacts with the tuberous sclerosis complex and promotes mTORC1 activity

**Pinhua Wang**[1†], **Sunayana Sarkar**[2†], **Menghuan Zhang**[1†], **Tingting Xiao**[1], **Fenhua Kong**[1], **Zhe Zhang**[1], **Deepa Balasubramanian**[2], **Nandan Jayaram**[3,4], **Sayantan Datta**[2‡], **Ruyu He**[1], **Ping Wu**[5], **Peng Chao**[5], **Ying Zhang**[6], **Michael Washburn**[6,7], **Laurence A Florens**[6], **Sonal Nagarkar-Jaiswal**[3,4], **Manish Jaiswal**[2]*, **Man Mohan**[1,8]*

[1]State Key Laboratory of Primate Biomedical Research, Institute of Primate Translational Medicine, Kunming University of Science and Technology, Kunming, China; [2]Tata Institute of Fundamental Research, Hyderabad, India; [3]CSIR–Centre for Cellular and Molecular Biology, Hyderabad, India; [4]Academy of Scientific and Innovative Research (AcSIR), Ghaziabad, India; [5]National Facility for Protein Science in Shanghai, Zhangjiang Lab, Shanghai, China; [6]Stowers Institute for Medical Research, Kansas City, United States; [7]Department of Cancer Biology, The University of Kansas Medical Center, Kansas City, United States; [8]Department of Biochemistry and Molecular Cell Biology, Shanghai Key Laboratory of Tumor Microenvironment and Inflammation, Shanghai Jiaotong University School of Medicine, Shanghai, China

**\*For correspondence:**
manish@tifrh.res.in (MJ);
manmohan2100@outlook.com
(MM)

[†]These authors contributed
equally to this work

**Present address:** [‡]Centre for
Writing and Pedagogy, School
of Interwoven Arts and Sciences,
Krea University, Sri City, India

**Competing interest:** The authors
declare that no competing
interests exist.

**Reviewing Editor:** Elizabeth P
Henske, Brigham And Women's
Hospital, United States

**Abstract** DYRK1A, a ubiquitously expressed kinase, is linked to the dominant intellectual developmental disorder, microcephaly, and Down syndrome in humans. It regulates numerous cellular processes such as cell cycle, vesicle trafficking, and microtubule assembly. DYRK1A is a critical regulator of organ growth; however, how it regulates organ growth is not fully understood. Here, we show that the knockdown of *DYRK1A* in mammalian cells results in reduced cell size, which depends on mTORC1. Using proteomic approaches, we found that DYRK1A interacts with the tuberous sclerosis complex (TSC) proteins, namely TSC1 and TSC2, which negatively regulate mTORC1 activation. Furthermore, we show that DYRK1A phosphorylates TSC2 at T1462, a modification known to inhibit TSC activity and promote mTORC1 activity. We also found that the reduced cell growth upon knockdown of DYRK1A can be rescued by overexpression of RHEB, an activator of mTORC1. Our findings suggest that DYRK1A inhibits TSC complex activity through inhibitory phosphorylation on TSC2, thereby promoting mTORC1 activity. Furthermore, using the *Drosophila* neuromuscular junction as a model, we show that the *mnb*, the fly homologs of *DYRK1A*, is rescued by RHEB overexpression, suggesting a conserved role of *DYRK1A* in TORC1 regulation.

## eLife assessment

This **fundamental** study identifies the kinase DYRK1A as a novel component of the tuberous sclerosis complex (TSC) protein complex, which is central to cellular growth and cell size. The findings presented here have broad implications for how cell size and growth is regulated. The methodology and analysis are **convincing** and support the findings.

## Introduction

The *dual-specificity tyrosine phosphorylation-regulated kinase 1 A (DYRK1A)* is a ubiquitously expressed kinase that belongs to the CMGC group (Cyclin-dependent kinases, Mitogen-activated protein kinase, Glycogen synthase kinases, and CDK-like kinases group). *DYRK1A,* is located within the Down syndrome critical region (DSCR). The phenotypes associated with Down syndrome (*Antonarakis, 2017*), including abnormal neurodevelopment (*Hämmerle et al., 2011*; *Najas et al., 2015*), and increased susceptibility to acute megakaryoblastic leukemia (*Malinge et al., 2012*), have been reasoned to be due to an increased expression of *DYRK1A*. On the other hand, haploinsufficiency of *DYRK1A* causes DYRK1A syndrome, a rare autosomal dominant disease characterized by intellectual disability, intrauterine growth retardation, microcephaly, and stunted growth (*Courcet et al., 2012*; *Kay et al., 2016*; *Møller et al., 2008*). Mouse models heterozygous for *Dyrk1a* also show a reduction in brain and body size (*Fotaki et al., 2002*; *Fotaki et al., 2004*). In *Drosophila*, loss of *mnb*, the *Drosophila* homolog of DYRK1A, results in reduced organ size, including brain, legs, and wings (*Degoutin et al., 2013*; *Tejedor et al., 1995*). Therefore, *DYRK1A* is a critical regulator of organ growth (*Guedj et al., 2012*); however, the cellular processes by which *DYRK1A* promotes organ growth are unclear.

*DYRK1A* and its homologs have also been implicated in many cellular processes, such as cell cycle, signal transduction, gene expression, vesicle trafficking, and microtubule assembly (*Arbones et al., 2019*). DYRK1A has been shown to extend the G1 phase of the cell cycle in neuroblastoma cells by phosphorylation and subsequent degradation of cyclin D1 (*Chen et al., 2013*). In glioblastoma cells, DYRK1A phosphorylates ubiquitin ligase CDC23, which mediates mitotic protein degradation, thereby promoting tumor growth (*Recasens et al., 2021*). Furthermore, DYRK1A expression positively correlates with EGFR levels, and DYRK1A inhibitors reduce EGFR-dependent glioblastoma growth (*Pozo et al., 2013*). DYRK1A promotes the expression of several genes required for ribosomal biogenesis and protein translation (*Di Vona et al., 2015*; *Li et al., 2018*); for example, RPS6 ribosomal protein S6, a component of the 40 S subunit, and translation initiation factor, EIF4A3 (*Xue et al., 2021*). However, if the regulation of these genes by DYRK1A is physiologically significant, it is currently unknown.

The mechanistic target of rapamycin complex 1 (mTORC1) is a key energy sensor and a master regulator of anabolic processes (*Rehbein et al., 2021*). The mTORC1 promotes anabolic processes such as protein, nucleotide, and lipid synthesis and represses catabolic processes such as autophagy, thereby promoting cellular growth (*Ben-Sahra and Manning, 2017*; *Saxton and Sabatini, 2017*). The mechanism of mTORC1-mediated regulation of multiple anabolic pathways has been studied extensively (*Rehbein et al., 2021*; *Ben-Sahra and Manning, 2017*; *Saxton and Sabatini, 2017*). TSC consisting of TSC1, TSC2, and TBC1D7 (TBC1 domain family member 7), integrates various extracellular signals and negatively regulates mTORC1 complex activity (*Rehbein et al., 2021*; *Dibble et al., 2012*). TSC2 is a GTPase-activating protein (GAP) and promotes the conversion of GTP-bound RHEB (Ras Homolog Enriched in Brain) to the GDP-bound state, thereby inhibiting it. When TSC2 is inhibited, the GTP-bound RHEB activates mTORC1 (*Li et al., 2003*; *Tee et al., 2003*). The mTORC1 activation is controlled by extracellular stimuli, nutrients, energy status, and stress. For example, in response to insulin stimulation, AKT phosphorylates TSC2 at several sites, including S939 and T1462, suppressing its inhibitory effect on RHEB-GTP recycling, thus activating mTORC1 (*Potter et al., 2002*). Similarly, in response to growth factors, extracellular signal-regulated Kinase (ERK) (*Ma et al., 2005*) and p90 ribosomal S6 kinase 1 (RSK1) phosphorylate TSC2 and activate mTORC1 (*Roux et al., 2004*). MK2 (MAP kinase-activated protein kinase 2/MAPKAPK2) phosphorylates TSC2 at yet another site, S1210, in the presence of serum and leads to the inhibition of TSC2 activity, which allows the activation of mTORC1 (*Li et al., 2003*). Catabolic signals, such as low energy levels during glucose scarcity, drive AMPK-mediated phosphorylation of TSC2 that promotes its GAP function and suppresses mTORC1 activity (*Corradetti et al., 2004*). Several kinases regulating the TSC activity underscores the importance of investigating novel kinases involved in its regulation, which may regulate mTORC1 activity in distinct cellular contexts.

In this study, we discover that DYRK1A interacts with TSC1 and TSC2. We show that DYRK1A phosphorylates TSC2 at T1462 and is a possible positive regulator of mTORC1 activity and cell growth. Furthermore, we show that in *Drosophila*, both Mnb, the DYRK1A-homolog, and mTORC1 are required to promote neuromuscular junction (NMJ) growth. Moreover, the activation of mTORC1 can

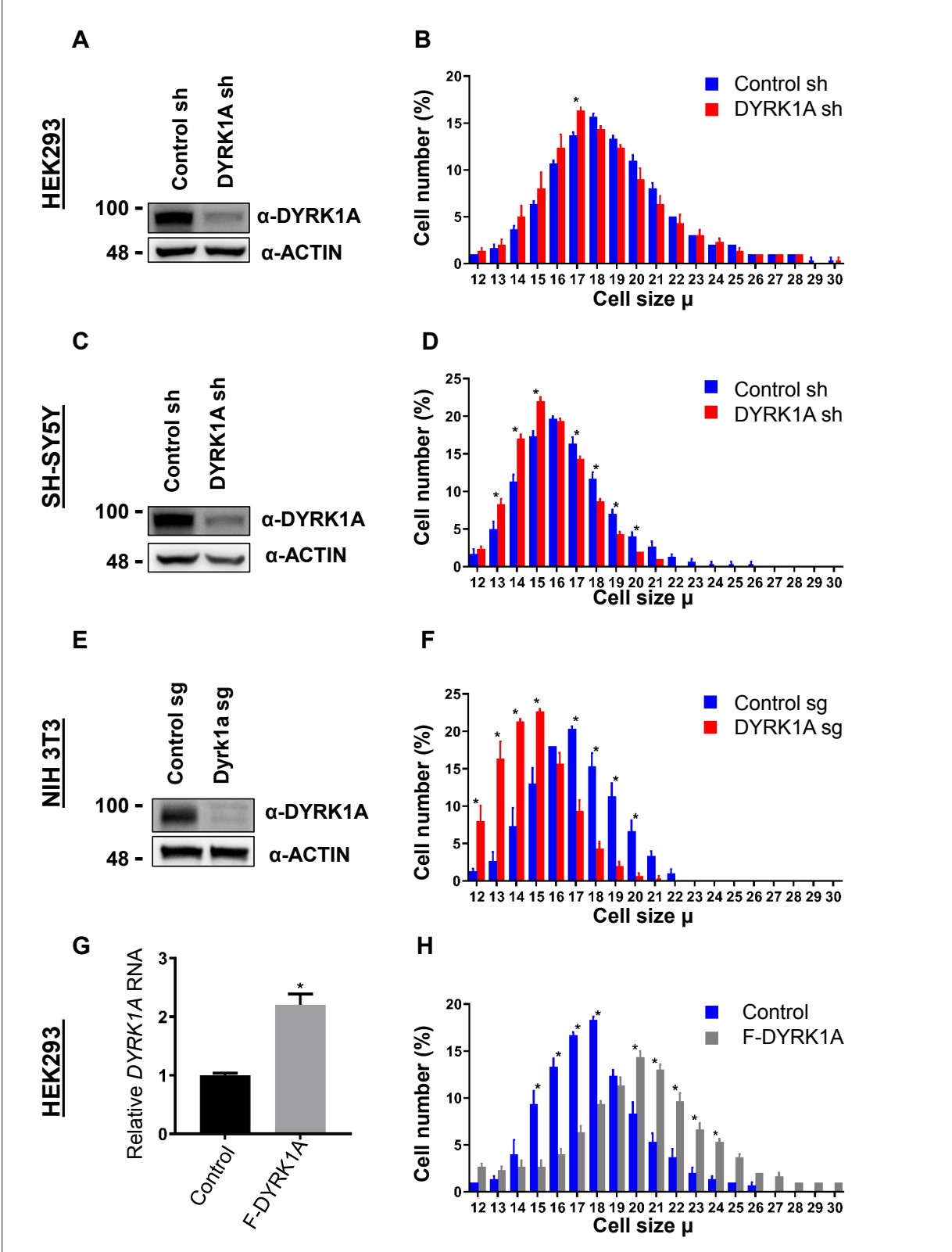

**Figure 1.** Dual-specificity tyrosine phosphorylation-regulated kinase 1A (DYRK1A) regulates cell size. shRNA-mediated knockdown of *DYRK1A* was performed in (**A, B**) HEK293 and (**C, D**) SH-SY5Y cells using lentivirus. Transduced cells were selected for four days before analysis. Western blot shows the efficiency of *DYRK1A* knockdown. (**E, F**) NIH3T3 cells were treated with *Dyrk1a*-targeting sgRNA expressing lentivirus and selected for four days before analysis. Western blot shows the efficiency of *DYRK1A* knockdown. (**G, H**) HEK293 cells expressing *Flag-DYRK1A* and the parental cells

*Figure 1 continued on next page*

*Figure 1 continued*

were treated with 40ng/ml Doxycycline for 48hr and analyzed for cell size. (**G**) Overexpression was analyzed by qRT-PCR. GAPDH mRNA was used to normalize RNA in q-RT-PCR samples. Data represent the mean ± SD (n=3 biological replicates).

The online version of this article includes the following source data and figure supplement(s) for figure 1:

**Source data 1.** Uncropped blots for *Figure 1A*.

**Source data 2.** Raw blots showing knockdown of DYRK1A for *Figure 1A*.

**Source data 3.** Uncropped blots for *Figure 1C*.

**Source data 4.** Raw blots showing knockdown of DYRK1A for *Figure 1C*.

**Source data 5.** Uncropped blots for *Figure 1E*.

**Source data 6.** Raw blots showing knockdown of Dyrk1a for *Figure 1E*.

**Figure supplement 1.** Analysis of cell size after induction of dual-specificity tyrosine phosphorylation-regulated kinase 1A (DYRK1A) expression with increasing dosage of Doxycycline.

rescue the NMJ growth phenotype of *mnb* loss of function, suggesting Mnb may regulate mTORC1 to promote neuromuscular junction morphology. We propose that DYRK1A-mediated regulation of mTORC1 activity is a conserved mechanism to regulate cell size and development.

## Results

### DYRK1A promotes cell growth

To study the function of *DYRK1A*, we knocked down DYRK1A in multiple cell lines, including in HEK293 cells (*Li et al., 2018*). We noticed that DYRK1A-depleted cells were smaller than the control cells. To explore if DYRK1A has a role in maintaining cell size, we performed a systematic analysis of the cell size phenotypes after transient knockdown of *DYRK1A* in HEK293 and observed a reduction in cell size (*Figure 1A and B*). Since mutations in *DYRK1A* are associated with stunted growth and microcephaly, we wondered if neuronal cells require *DYRK1A* to maintain proper cell size. Therefore, we knocked down *DYRK1A* in the human neuroblastoma cell line SH-SY5Y and observed a significant reduction in cell size (*Figure 1C and D*). Furthermore, we tested if the observed cell size phenotype is conserved in mice. We targeted *Dyrk1a* in NIH-3T3 using CRISPR/Cas9 and observed a significant reduction in cell size (*Figure 1E and F*). Our data suggests that *DYRK1A* promotes cell size across cell lines derived from different models and tissues. Thus, we tested if the overexpression of *DYRK1A* can promote cell size. Since strong overexpression of *DYRK1A* leads to cell cycle exit (*Hämmerle et al., 2011*; *Litovchick et al., 2011*; *Soppa et al., 2014*), we used the Doxycycline-inducible system to express low levels of Flag-DYRK1A in HEK293 cells (*Zhang et al., 2021*). We noted a change in cell size with an increasing dosage of Doxycycline, which reached a saturation state at 100 ng/ml Doxycycline in our system (*Figure 1—figure supplement 1*). We observed a significant increase in cell size at 40 ng/ml Doxycycline treatment for 48 hr when the DYRK1A mRNA levels were approximately twice that of control (*Figure 1G and H*). At this level of DYRK1A mRNA expression, overexpressed Flag-DYRK1A was undetectable by western blot. Overall, our knockdown and overexpression experiments in different cell lines, including both human and mouse, suggest that *DYRK1A* promotes cell size.

### DYRK1A physically interacts with the tuberous sclerosis complex

To understand how *DYRK1A* promotes cell growth, we investigated DYRK1A protein-interactors. Previously, we have identified protein-interactors of wild-type and kinase-dead DYRK1A (K188R) from HEK293 cells using affinity purification and mass spectrometry (*Li et al., 2018*; *Zhang et al., 2021*). In these purifications, we have observed significant enrichment of DCAF7, ARIP4, RB1, EP300, CREBBP, TRAF2, TRAF3, FAM53C, and RNF169, many of which have also been identified by many other groups as bona fide DYRK1A interacting proteins (*Guard et al., 2019*; *Li et al., 2021*; *Menon et al., 2019*; *Miyata and Nishida, 2011*; *Sitz et al., 2004*). We also observed significant enrichment of TSC components TSC1 and TSC2 among proteins co-purified with wild-type and kinase-dead DYRK1A (K188R) (*Figure 2A*). Interestingly, a few peptides of TBC1D7, the third component of TSC, were also detected in the wild-type pull-down (see dNSAF values in *Figure 2A*), suggesting that DYRK1A interacts with

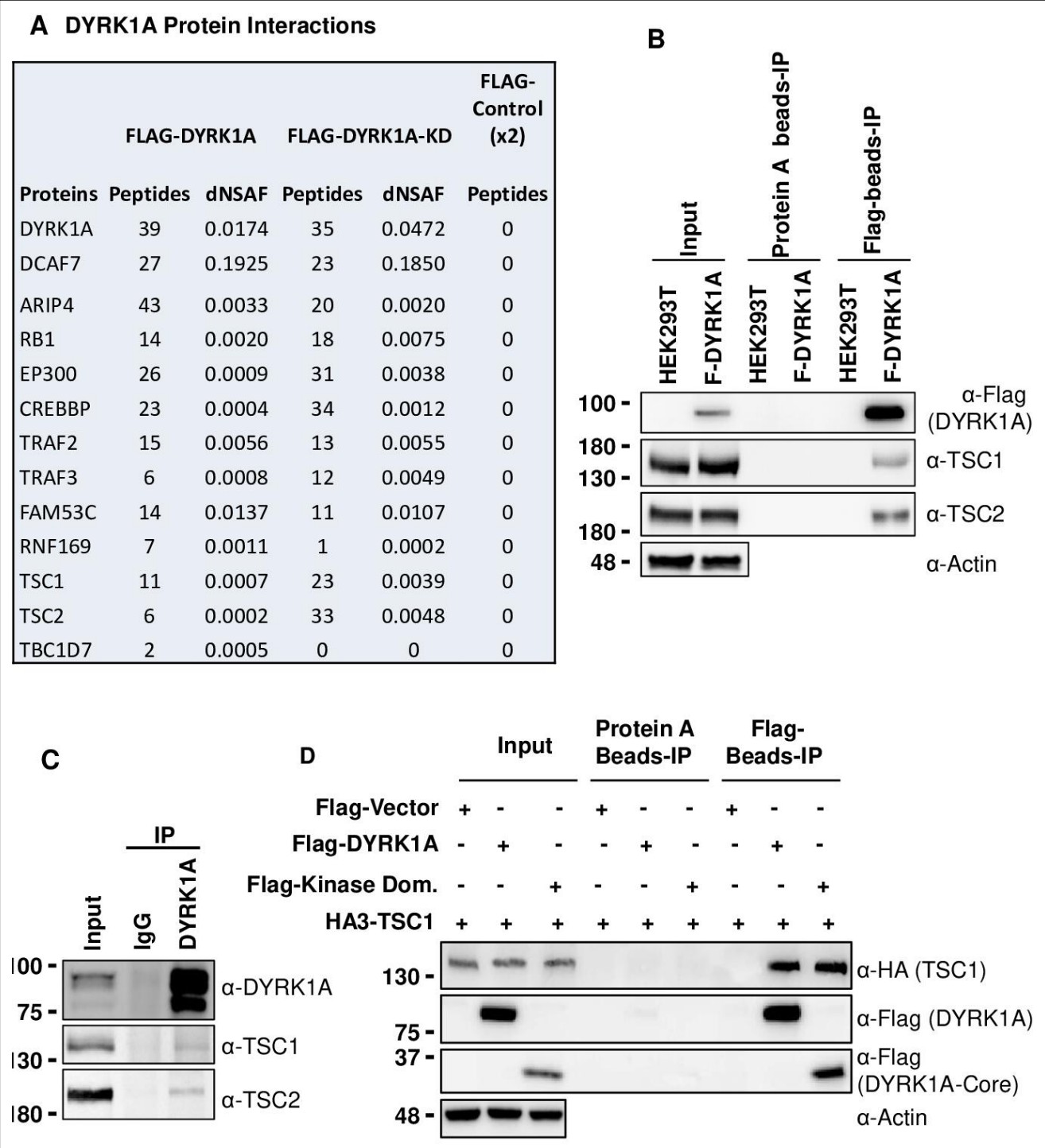

**Figure 2.** Dual-specificity tyrosine phosphorylation-regulated kinase 1A (DYRK1A) interacts with the tuberous sclerosis complex (TSC). (**A**) The tandem mass spectrometry (MS/MS) datasets previously acquired by MudPIT analyses of FLAG-DYRK1A affinity purifications and negative FLAG controls (*Li et al., 2018*) were searched against the most recent releases of the human protein sequence databases (built by collating and removing redundant entries from NCBI *Homo sapiens* RefSeq GCF_000001405.40_GRCh38.p14 and GCF_009914755.1_T2T-CHM13v2.0). Highly enriched proteins include known and novel DYRK1A-interacting partners and are reported with their peptide counts and distributed normalized spectral abundance factor (dNSAF) values, which reflect their relative abundance in the samples (*Zhang et al., 2010*). (**B**) Flag beads were used to pull down Flag-DYRK1A from whole cell extracts of HEK293 transfected with Flag-DYRK1A, and Protein A beads were used as a control. The blots were probed with TSC1 and TSC2 antibodies. Actin was used to normalize the lysate inputs. (**C**) Endogenous DYRK1A was immunoprecipitated with DYRK1A antibody from HEK293 cytoplasmic fraction generated using the Dignam protocol (*Li et al., 2018*) and probed with antibodies against endogenous DYRK1A, TSC1, and TSC2.

*Figure 2 continued on next page*

*Figure 2 continued*

Rabbit IgG was used as the IP control (**D**) Flag-DYRK1A and Flag-DYRK1A kinase domain constructs were affinity purified using Flag-beads from HEK293 cells co-transfected with HA3-TSC1 and probed with α-HA and α-Flag antibodies. Actin was used as the loading control.

The online version of this article includes the following source data and figure supplement(s) for figure 2:

**Source data 1.** Uncropped blots for *Figure 2B*.

**Source data 2.** Raw data of Flag-DYRK1A affinity purification and Flag, TSC1 and TSC2 blot for *Figure 2B*.

**Source data 3.** Uncropped blots for *Figure 2C*.

**Source data 4.** Raw data of interaction between endogenous DYRK1A and TSC1 and TSC2 for *Figure 2C*.

**Source data 5.** Uncropped blots for *Figure 2D*.

**Source data 6.** Raw data of interaction between Kinase domain of Flag-DYRK1A and HA-TSC1 for *Figure 2D*.

**Figure supplement 1.** TSC1/TSC2 interact with dual-specificity tyrosine phosphorylation-regulated kinase 1A (DYRK1A).

**Figure supplement 1—source data 1.** Uncropped blots for *Figure 2—figure supplement 1*.

**Figure supplement 1—source data 2.** Raw data of interaction between overexpressed TSC1 and TSC2 with endogenous DYRK1A for *Figure 2—figure supplement 1*.

**Figure supplement 2.** Dual-specificity tyrosine phosphorylation-regulated kinase 1A (DYRK1A) kinase domain interacts with TSC1.

**Figure supplement 2—source data 1.** Uncropped blots for *Figure 2—figure supplement 2*.

**Figure supplement 2—source data 2.** Raw data of interaction between truncated DYRK1A and full length-HA-TSC1 for *Figure 2—figure supplement 2*.

the TSC complex. This interaction is intriguing as TSC is a negative regulator of the mTORC1, which is known to promote cell growth (*Astrinidis et al., 2002*; *Gao and Pan, 2001*; *Tapon et al., 2001*).

To ascertain the interaction between DYRK1A and TSC, we expressed *Flag-DYRK1A* in HEK293 cells through the Doxycycline-inducible system and performed Flag affinity purification. Surprisingly, when we used HEPES buffer containing either 0.5% NP40 or 1% Tween 20, we did not detect co-purification of TSC2 with Flag-DYRK1A (data not shown). However, when we prepared extracts by lysing the cells in hypotonic buffer without detergent followed by Dounce homogenization, we found that both TSC1 and TSC2 co-purify with Flag-DYRK1A (*Figure 2B*). Furthermore, we tested the interaction between endogenous DYRK1A and TSC by immunoprecipitating DYKR1A; we observed co-purification of both TSC1 and TSC2 (*Figure 2C*). Immuno-affinity purification of overexpressed HA3-TSC1 and Flag-TSC2 was also able to pull down endogenous DYRK1A (*Figure 2—figure supplement 1*). These results confirmed that DYRK1A physically interacts with the TSC complex. We then sought to identify the domains of DYRK1A that interact with TSC1, as TSC1 is the scaffolding protein in the TSC (*Rehbein et al., 2021*). We co-overexpressed HA3-TSC1 and various truncated forms of Flag-DYRK1A in HEK293 cells (*Figure 2—figure supplement 2*). We found that interaction between DYRK1A and TSC1 was abolished in all the truncated forms that lack the DYRK1A kinase domain, which suggests that the DYRK1A kinase domain is required for the interaction (*Figure 2—figure supplement 2*). We further confirmed that DYRK1A interacts with TSC1 through its kinase domain by co-expressing the FLAG-tagged DYRK1A kinase domain and HA3-TSC1, followed by immunoprecipitation using anti-FLAG antibodies (*Figure 2D*). Taken together, our results show that DYRK1A physically interacts with the TSC.

## DYRK1A promotes mTORC1 activity

Next, we sought to test the implications of TSC and DYRK1A interaction on mTORC1 activity. mTORC1 is known to be induced in HEK293 cells by treatment with fetal bovine serum (FBS), insulin, and many growth factors following serum starvation (*Ben-Sahra and Manning, 2017*; *Saxton and Sabatini, 2017*; *Inoki et al., 2003*; *Wu and Storey, 2021*). We performed similar experiments, where we knocked down *DYRK1A* followed by serum starvation and tested for levels of pS6K, a reporter for mTOR activity. We did not observe a significant reduction in basal phosphorylation levels on S6K and S6. However, 10 and 20 min after FBS treatment, we observed significantly reduced levels of Phospho-S6K and Phospho-S6 in *DYRK1A* KD compared to control HEK293 cells (*Figure 3A–C*). Furthermore, we tested the levels of pS6K and pS6 in NIH3T3 after CRISPR-mediated targeting of *Dyrk1a*. Interestingly, we found significant reductions in pS6K and pS6 compared to controls (*Figure 3D–F*). Overall, our results indicate that DYRK1A is required to promote mTORC1 activity.

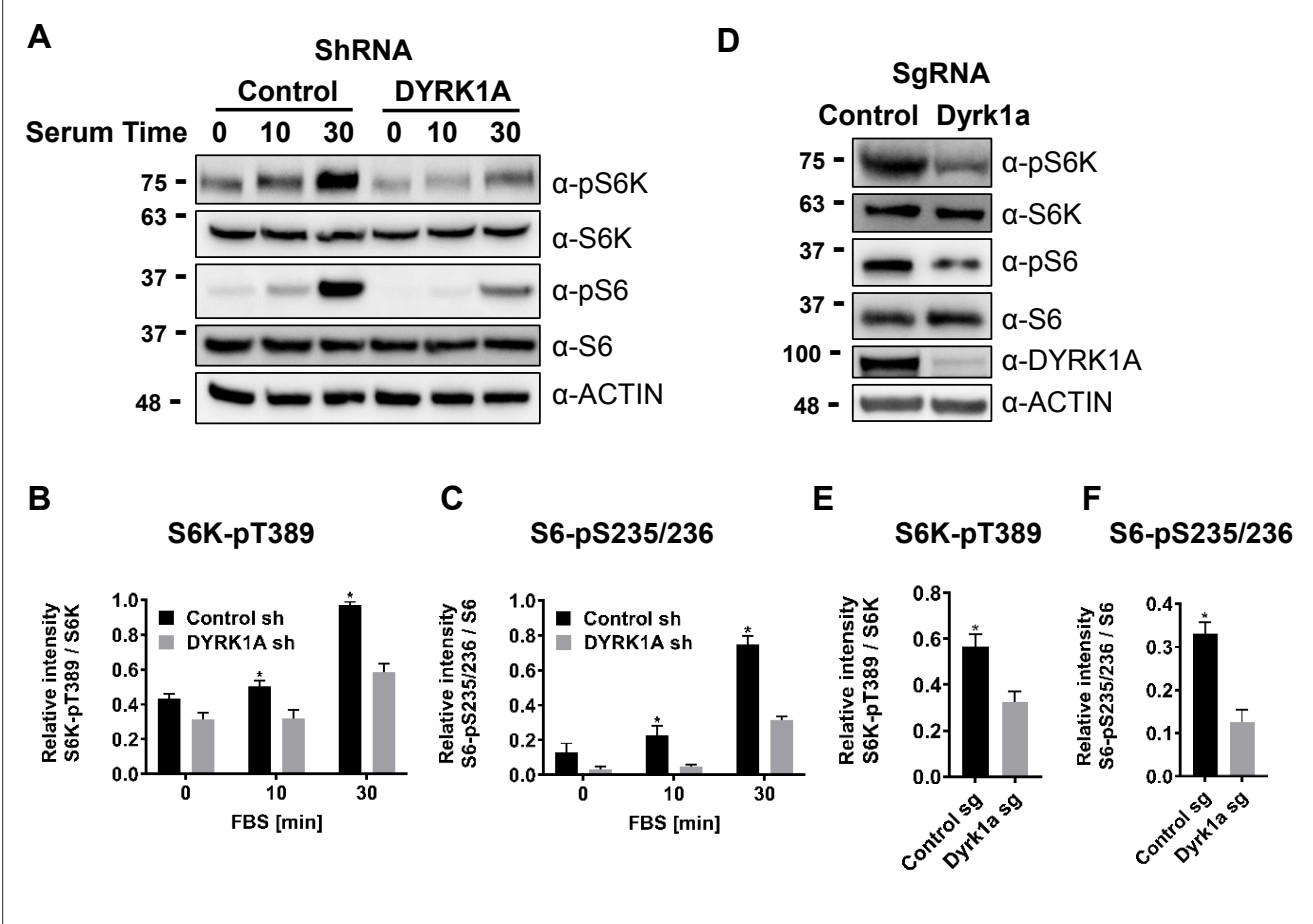

**Figure 3.** Dual-specificity tyrosine phosphorylation-regulated kinase 1A (DYRK1A) promotes the activation of mTORC1 pathway in human and mouse cells. (**A**) HEK293 cells treated with *DYRK1A* short hairpin RNA (shRNA) or control shRNA were serum starved for 12 hr before being activated with serum for the indicated times. Cells were then harvested, lysates, and probed with the indicated antibodies. Actin was used as the loading control. (**B, C**) Quantification of proteins in (**A**), levels of pS6K (T389), S6K, pS6 (pS235/236), and S6 were quantified using Image J software and the ratio of pS6K/S6K and pS6/S6 were plotted (n=3 biological replicates). (**D**) NIH3T3 cells were treated with sgRNA-targeting *Dyrk1a* or non-targeting control and selected for four days with Puromycin before harvesting. Lysates were probed with indicated antibodies. (**E, F**) Quantification of proteins in (**D**), levels of pS6K (T389), S6K, pS6 (pS235/236), and S6 were quantified (as described for B and C) and ratios were plotted (n=3 biological replicates). Student's *t*-tests were done to compare samples. p-value = *p<0.05.

The online version of this article includes the following source data for figure 3:

**Source data 1.** Uncropped blots for *Figure 3A*.

**Source data 2.** Raw data showing phosphorylation status of S6k and S6 after knockdown of DYRK1A for *Figure 3A*.

**Source data 3.** Uncropped blots for *Figure 3D*.

**Source data 4.** Raw data showing phosphorylation status of S6K and S6 after CRISPR knockout of Dyrk1a in NIH3T3 cells for *Figure 3D*.

## DYRK1A phosphorylates TSC2 at T1462

Next, we investigated if DYRK1A phosphorylates TSC1/TSC2, which is known to regulate mTORC1 activity. For this, we first knocked down *DYRK1A* in HEK293 cells, co-expressed *HA3-TSC1* and *Flag-TSC2,* and performed immunoprecipitation followed by LC/tandem MS (MS/MS) to identify the phosphorylation status of TSC1 and TSC2 in these cells. We observed that *DYRK1A* knockdown samples have reduced TSC2 phosphorylation at T1462 (T1462) compared to the control samples. T1462 has been previously identified as a site phosphorylated by various kinases that activate the mTORC1 pathway, including AKT (*Potter et al., 2002*), ERK (*Ma et al., 2005*), p90 RSK1 (*Roux et al., 2004*), IκB kinase β (IKKβ) (*Manning et al., 2002*), and MAPKAPK2 (MK2) (*Li et al., 2003*) in response to insulin and other growth factors. To ascertain that indeed, TSC2 T1462 is the

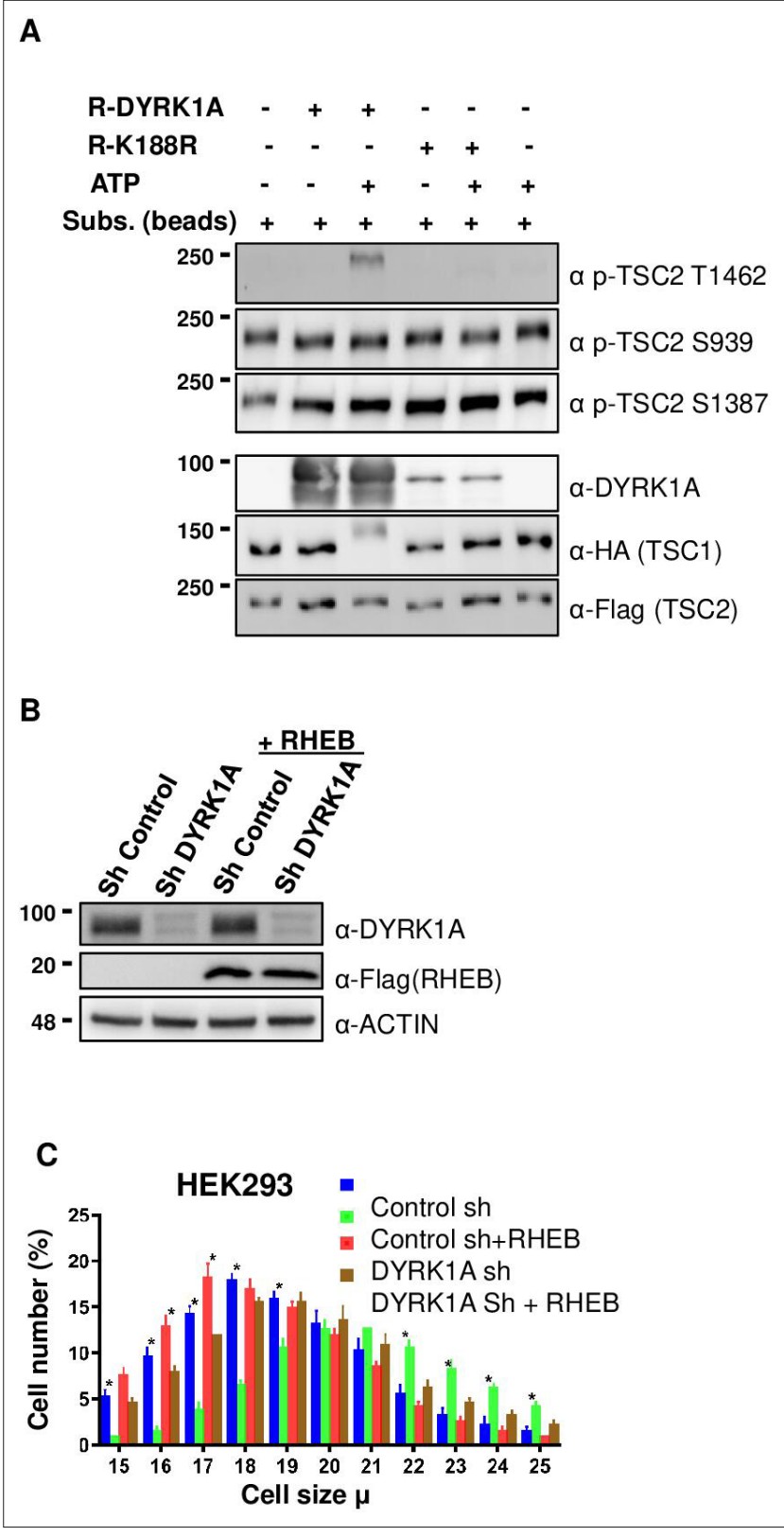

**Figure 4.** Dual-specificity tyrosine phosphorylation-regulated kinase 1A (DYRK1A) phosphorylates TSC2 at T1462 in vitro, and Ras Homolog Enriched in Brain (RHEB) overexpression rescues mTORC1 activity in cells. (**A**) An in-vitro kinase assay was performed using DYRK1A and kinase-dead DYRK1A (K188R) that were purified from bacteria. Flag-TSC2 and HA3-TSC1 were co-expressed in HEK293 cells and purified using a combination of (1:1) of HA and

*Figure 4 continued on next page*

*Figure 4 continued*

Flag beads. Beads were equilibrated with kinase assay buffer before the reactions were initiated on beads. After incubation for 30 min at 30°C, reactions were stopped by the addition of SDS loading buffer. Since bacterially purified DYRK1A is autophosphorylated, it exhibits a fuzzier signal, whereas kinase-dead DYRK1A is incapable of phosphorylation and appears as a sharp signal. (**B, C**) *RHEB* overexpression partially rescues the size of HEK293 cells. HEK293 cells were first transduced with short hairpin RNA (shRNA) lentivirus targeting DYRK1A or control and selected with 1 ug/ml Puromycin for three days, after which they were re-transduced with lentivirus expressing Flag-RHEB. The concentration of Puromycin was raised to 2 ug/ml for the next 48 hr in order to select for the second round of transduction. (**B**) Panel shows knockdown efficiency of DYRK1A and overexpression of RHEB. (**C**) Lower panel shows cell size analysis. Data represent the mean ± SD (n=3 biological replicates). Student's t-test was done to compare samples. Significant difference in p-value = *p<0.05.

The online version of this article includes the following source data and figure supplement(s) for figure 4:

**Source data 1.** Uncropped blots for *Figure 4A*.

**Source data 2.** Raw data showing in vitro kinase assay using recombinant DYRK1A and K188R for *Figure 4A*.

**Source data 3.** Uncropped blots for *Figure 4B*.

**Source data 4.** Raw data showing knockdown of DYRK1A and overexpression of Flag-RHEB for *Figure 4B*.

**Figure supplement 1.** mTORC1 inhibitors block the increase in cell size mediated by dual-specificity tyrosine phosphorylation-regulated kinase 1A (DYRK1A).

site phosphorylated by DYRK1A, we performed in-vitro kinase assays using purified wild-type and kinase-dead DYRK1A. As TSC2 is about 200 KDa protein, we purified transiently overexpressed HA3-TSC1 and Flag-TSC2 from HEK293 cells and performed in vitro kinase assay on beads. Wild-type DYRK1A autophosphorylates and runs as fuzzy bands around 90–100 Kda in polyacrylamide gels, whereas kinase-dead DYRK1A runs only as a single sharp band (*Figure 4A*). In our assay, we did not observe an increase in phosphorylation at S1387 or Serine 939 in the presence of ATP and DYRK1A (*Figure 4A*). However, immunopurified TSC2 is heavily phosphorylated at Serine 1387 and Serine 939, which are two important sites that are phosphorylated by AKT (*Potter et al., 2002*; *Inoki et al., 2003*; *Manning et al., 2002*). Therefore, we cannot rule out if the immunopurified TSC2 was fully phosphorylated at S1387 and S939. However, it is important to note that we did not observe any differential loss of phosphorylation in TSC2 at S1387 in our mass spectrometry phosphorylation analysis. Interestingly, we found that T1462 was strongly phosphorylated by wild-type DYRK1A, not by kinase-dead DYRK1A (*Figure 4A*). Taken together, these experiments demonstrate that DYRK1A phosphorylates TSC2 on T1462. This phosphorylation is known to be an inhibitory modification and it is a key site that gets phosphorylated by various other key regulators of the mTORC1 pathway (*Manning et al., 2002*). We could not perform in-vitro tests to check if DYRK1A phosphorylates TSC1 or other sites on TSC2 due to the lack of appropriate antibodies. The overall data suggest that DYRK1A mediates TSC2 T1462 phosphorylation to inhibit TSC activity, promoting TORC1 activity.

## RHEB overexpression can rescue cell growth phenotype of *DYRK1A* knockdown

TSC phosphorylation has been shown to inhibit TSC, and activate mTORC1 via RHEB, where the activation of TSC leads to the hydrolysis of GTP-bound RHEB, which in turn leads to mTORC1 inhibition (*Rehbein et al., 2021*). We hypothesized that DYRK1A-mediated phosphorylation of TSC2 may promote mTORC1 activation in a RHEB-mediated manner. Therefore, we reasoned that overexpression of *RHEB* will rescue the cell size phenotype caused by *DYRK1A* knockdown. We overexpressed *RHEB* in *DYRK1A* knockdown cells and found a partial rescue of cell size, suggesting that DYRK1A indeed plays a role in cell size regulation at least partly upstream to RHEB (*Figure 4B and C*). Furthermore, inhibition of mTOR with either Rapamycin or Torin1 rescued the increased cell size phenotype mediated by overexpression of *DYRK1A* in the inducible HEK293A-Flag-DYRK1A model (*Figure 4—figure supplement 1*). These data suggest that DYRK1A phosphorylates TSC2, which inhibits its action, leading to the activation of mTORC1 via RHEB.

## *Drosophila mnb* mutants show reduced neuromuscular junctions, which can be rescued by RHEB overexpression

Since the TSC-TOR pathway is conserved in *Drosophila* (*Takahara and Maeda, 2013*), we sought to test if the DYRK1A-mediated regulation of the TSC-TOR pathway is also conserved. The TOR pathway is shown to promote NMJ development in flies (*Dimitroff et al., 2012*; *Natarajan et al., 2013*; *Figure 5B and G*). Similarly, *mnb*, the fly homolog of DYRK1A, is also required for NMJ growth (*Chen et al., 2014*); however, the mechanism of *mnb*-mediated NMJ growth is unknown. Based on our finding that *DYRK1A* promotes TOR activity, we hypothesized that *mnb* promotes larval NMJ growth through TOR. To test this hypothesis, we systematically compared the NMJ growth phenotype of *mnb* mutant with that of loss or gain of TOR activity at larval NMJ. Since TOR mutants are early larval lethal, we expressed dominant-negative TOR (*TOR.Ted*) in motor neurons using the UAS-Gal4 system (*D42Gal4>UAS TOR.Ted*) (*Brand and Perrimon, 1993*). To activate the TOR pathway, we overexpressed *RHEB* in motor neurons (*D42Gal4>UAS* RHEB) (*Gustafson and Boulianne, 1996*). We stained NMJ using anti-HRP (to mark neurons) and anti-DLG (to mark boutons where synaptic connections are formed). As shown in *Figure 5A, B and G*, the loss of *mnb* causes a reduction in the NMJ bouton numbers compared to the wild-type control (WT, Canton S)—similarly, the expression of TOR.TED in motor neurons (*D42Gal4>UAS Tor.Ted*) also causes a reduction in NMJ bouton numbers (*Figure 5—figure supplement 1D-F*). In contrast, overexpression of *mnb* in motor neurons increases the bouton numbers (*Figure 4C, D and H*). Similarly, the activation of the TOR pathway by overexpression of *RHEB* in motor neurons or mutations in *gigas* (*gig*, the fly homolog of *TSC2*) increases the number of boutons (*Figure 5E, F, I, Figure 5—figure supplement 1C*). The similarities in the TOR and *mnb* phenotypes suggest that Mnb may positively regulate TOR activity and promote NMJ growth. To test this hypothesis, we asked whether or not the NMJ phenotype of the *mnb* mutant is rescued by the activation of TOR. As shown in *Figure 5L*, overexpression of *RHEB* rescues the NMJ bouton numbers in *mnb* mutants. These results indicate that *RHEB* functions downstream to *mnb* to regulate the NMJ development. Taken together, the results in mammalian cells and flies suggest that the role of DYRK1A/Mnb in regulating the TOR pathway is conserved.

## Discussion

This work shows that DYRK1A promotes cell size as a reduction in *DYRK1A* levels in both human and mouse cells results in small cell size (*Figure 1A–D*). mTORC1 is a key energy sensor and a master regulator of cell growth (*Ben-Sahra and Manning, 2017*; *Saxton and Sabatini, 2017*). We found that reduced levels of *DYRK1A* in cells result in decreased S6K and S6 phosphorylation, suggesting that DYRK1A positively regulates mTORC1 (*Figure 3*). One of the key regulators of mTORC1 is the TSC complex, consisting of TSC1, TSC2, and TBC1D7 (TBC1 domain family member 7), which integrates various extracellular signals and negatively regulates mTORC1 complex activity (*Rehbein et al., 2021*; *Dibble et al., 2012*). DYRK1A physically interacts with the TSC complex through its kinase domain (*Figure 2*). TSC2 is a GTPase-activating protein (GAP) that promotes the conversion of GTP-bound RHEB to the GDP-bound state, thereby inhibiting it. When TSC2 is inhibited, the GTP-bound RHEB activates mTORC1 (*Tee et al., 2003*; *Inoki et al., 2003*). Phosphorylation at TSC2 at T1462 has been shown to inhibit TSC and promote TORC1 activity through RHEB (*Manning et al., 2002*; *Zhang et al., 2009*). Our study, for the first time, shows that DYRK1A phosphorylates TSC2 at the T1462 site, thereby promoting the mTORC1 pathway. We also show that the overexpression of *RHEB* could partially rescue the cell growth phenotypes exhibited by *DYRK1A* knockdown cells, indicating that the DYRK1A works upstream to RHEB.

In DYRK1A syndrome, the loss of one copy of *DYRK1A l*eads to stunted growth and microcephaly in humans (*Møller et al., 2008*). A recent report showing a reduction in mTOR signaling in mice cortex lacking *Dyrk1a* indicates that the DYRK1A-mTORC1 axis may regulate brain development (*Levy et al., 2021*). In *Drosophila*, *mnb* is widely expressed in the developing central nervous system and has been linked to post-differentiation roles in neurons. For example, in flies, *mnb* has been linked to long-term memory formation (*Kacsoh et al., 2019*). In humans and mice, *DYRK1A* has been linked to the neurite spine morphogenesis, and it has been suggested that the neurite spine phenotype may contribute to intellectual disability in Down syndrome (*Haas et al., 2013*; *Martinez de Lagran et al., 2012*; *Park and Chung, 2013*). The phenomenon of NMJ morphogenesis in fruit flies is analogous

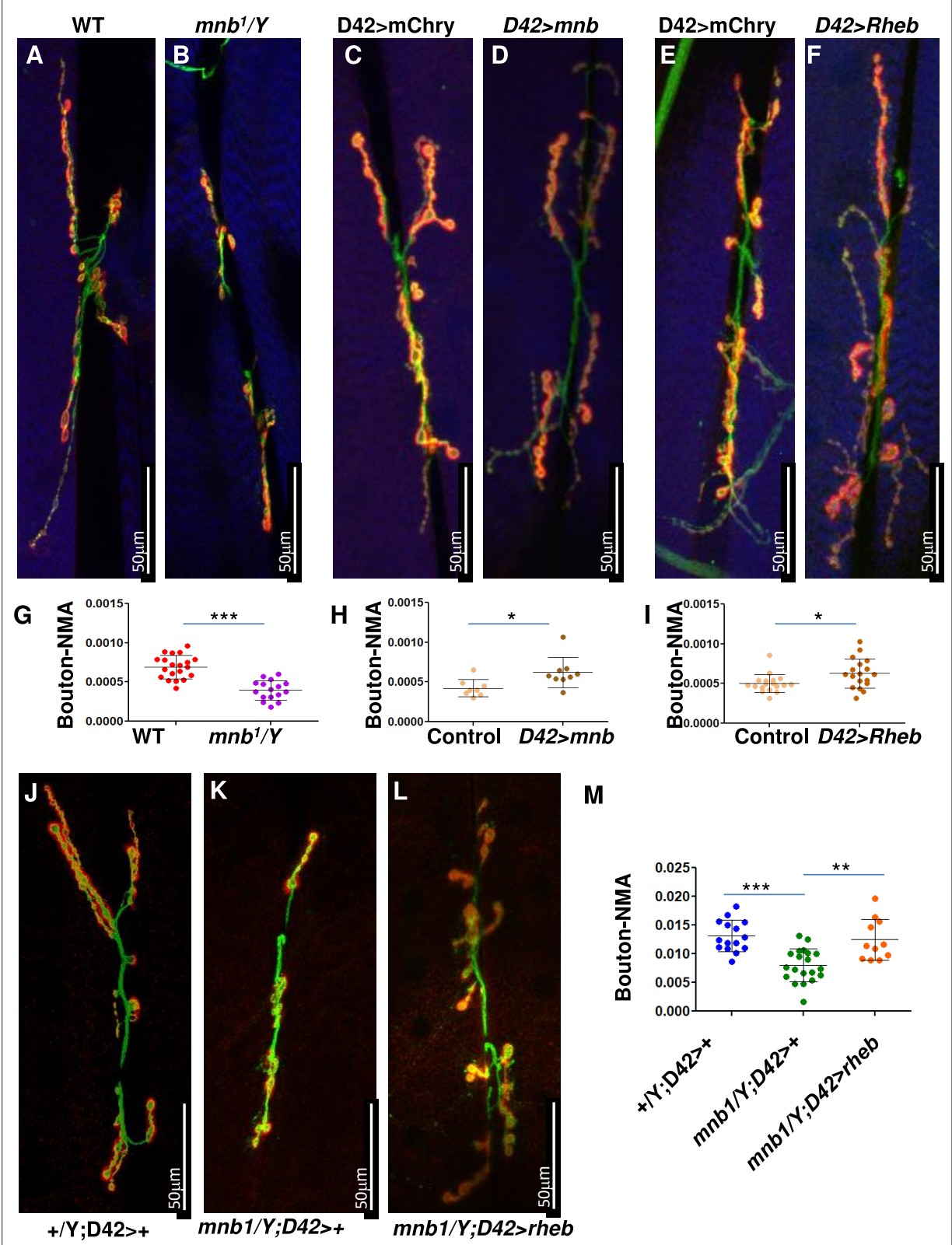

**Figure 5.** *mnb* mutant phenotype can be rescued by TOR activation in flies. (**A–F and J–L**) Third instar larval neuromuscular junction (NMJ) (muscles 6/7) were stained using anti-HRP (Green) and anti-Dlg (Red). Muscles are stained with phalloidin (Blue, **A–F**). HRP (green) stains the entire neuron and Dlg (red) stains only boutons (Red + Green). (**G–I, M**) Quantification of bouton numbers, normalized to muscle area (Bouton-NMA). Error bars represent standard deviation. Statistical significance (p-values: ***<0.001; **<0.01; *<0.05) is calculated by unpaired student's t-test. (**A, B, G**) *mnb¹* alleles show

*Figure 5 continued*

fewer boutons numbers as compared to wild-type (WT, Canton S) control (**B**). Data are quantified in G. (**C, D, H**) *mnb* overexpression (*D42-Gal4*>UAS *mnb*, **D**) increases bouton numbers as compared with mCherry overexpression (*D42-Gal4*>UAS-mCherry, Control, **C**). D42-Gal4 is a motor-neuron-specific driver. Data are quantified in H. (**E, F, I**) Rheb overexpression (*D42-Gal4*>UAS Rheb, **F**) increases bouton numbers as compared with mCherry overexpression (*D42-Gal4*>UAS-mCherry, Control, **E**). Data are quantified in I. (**J–M**) *Rheb* overexpression in *mnb* mutant (*mnb[1]/Y D42-Gal4*>UASRheb, **L**) suppressed bouton phenotype as compared to *mnb* mutant (*mnb[1]/Y D42-Gal4/+*, **K**). Wild-type is heterozygous *D42-Gal4* (*+/Y; D42-Gal4/+*, **J**). Data is quantified in (**M**).

The online version of this article includes the following figure supplement(s) for figure 5:

**Figure supplement 1.** Neuromuscular junction (NMJ) phenotypes due to TOR gain or loss.

to neurite spine morphogenesis. The TOR pathway is shown to promote NMJ development in flies (*Dimitroff et al., 2012*; *Natarajan et al., 2013*; *Figure 5B and G*). Similarly, *mnb* is also required for NMJ growth (*Chen et al., 2014*); however, the mechanism of *mnb*-mediated NMJ growth was unknown. Our results show that the reduction in NMJ size in *mnb* mutants can be rescued by *RHEB* overexpression, suggesting that Mnb positively regulates mTORC1. Our results are also supported by a study in *Arabidopsis* that showed YAK1, a DYRK kinase, regulates meristem activity through the mTOR pathway (*Barrada et al., 2019*). Overall, our results suggest that the role of Mnb/DYRK1A in the regulation of the mTORC1 pathway is evolutionarily conserved, and the DYRK1A-mTORC1 axis plays a crucial role in neuronal morphogenesis and cellular growth.

# Methods

## Key resources table

| Reagent type (species) or resource | Designation | Source or reference | Identifiers | Additional information |
|---|---|---|---|---|
| Strain (*Drosophila melanogaster*) | *Canton S* | Bloomington Stock Center | | |
| Strain (*D. melanogaster*) | *mnb[1]* | *Tejedor et al., 1995* | FBal0012364 | gift from Francisco J. Tejedor |
| Genetic reagent (*D. melanogaster*) | UAS-*mnb* | *Shaikh et al., 2016* | FBtp0114512 | gift from Francisco J. Tejedor |
| Genetic reagent (*D. melanogaster*) | UAS-*RHEB* | Bloomington Stock Center | FBst0009688 | |
| Genetic reagent (*D. melanogaster*) | D42-Gal4 | Bloomington Stock Center; *Gustafson and Boulianne, 1996* | FBti0002759 | |
| Genetic reagent (*D. melanogaster*) | P[(45)w[+mC]=UAS Tor.TED]II | Bloomington Stock Center; *Shaikh et al., 2016* | FBti0026636 | |
| Genetic reagent (*D. melanogaster*) | UAS-mCherry | Bloomington Stock Center | FBti0147460 | |
| Chemical compound | 4% paraformaldehyde | Himedia | Cat# TCL119 | |
| Antibody | Mouse anti-DLG | DSHB; *Shaikh et al., 2016* | CatID# 4F3 | IF (1:500) |
| Antibody | Rabbit anti-HRP conjugated with alexa488 | Jackson | CatID# 123-545-021 | IF (1:500) |
| Antibody | Goat anti-mouse conjugated with Alexa 555 | Invitrogen | CatID# A28180 | IF (1:500) |
| Other | Microsocope: Leica Stellaris 5 | Leica | | PL APO 40 X/1.30 oil objective |
| Other | Microsocope: Olympus FV3000 | Olympus | | UPLFLN 40 X/1.30 oil objective |
| Cell line (*Homo sapiens*) | HEK293 | ATCC | CRL-1573 | |
| Cell line (*H. sapiens*) | 293T | ATCC | CRL-3216 | |
| Cell line (*Mus musculus*) | NIH3T3 | ATCC | CRL-1658 | |
| Cell line (*H. sapiens*) | SH-SY5Y | ATCC | CRL-2266 | |

*Continued on next page*

*Continued*

| Reagent type (species) or resource | Designation | Source or reference | Identifiers | Additional information |
|---|---|---|---|---|
| Transfected construct (*H. sapiens*) | DYRK1A shRNA | ThermoFisher; *Li et al., 2018* | | Lentiviral construct to transduce and express the shRNA. |
| Transfected construct (*M. musculus*) | *Dyrk1a* sgRNA | This paper | | Lentiviral construct to transduce and mediate Dyrk1a knockout |
| Antibody | anti-Actin (rabbit monoclonal) | Abclonal | Cat# AC026 | WB (1:5000) |
| Antibody | anti-HA (mouse monoclonal) | Abclonal | Cat# AE008 | WB (1:5000) |
| Antibody | anti-TSC1 (rabbit polyclonal) | Cell Signaling Technology | Cat# 4906 | WB (1:1000) |
| Antibody | anti-TSC2 (rabbit monoclonal) | Cell Signaling Technology | Cat# 4308 | WB (1:1000) |
| Antibody | anti- p70 S6 Kinase (rabbit monoclonal) | Cell Signaling Technology | Cat# 2708 | WB (1:1000) |
| Antibody | anti- Phospho-p70 S6 Kinase (Thr389) (rabbit polyclonal) | Cell Signaling Technology | Cat# 9205 | WB (1:1000) |
| Antibody | anti- S6 Ribosomal Protein (mouse monoclonal) | Cell Signaling Technology | Cat# 2317 | WB (1:1000) |
| Antibody | anti- Phospho-S6 Ribosomal Protein (Ser235/236) (rabbit monoclonal) | Cell Signaling Technology | Cat# 4856 | WB (1:1000) |
| Antibody | anti-Flag(mouse monoclonal) | MBL | Cat# M185 | WB (1:5000) |
| Antibody | anti-Phospho-TSC2-T1462(rabbit polyclonal) | Abclonal | Cat# AP0866 | WB (1:1000) |
| Antibody | anti-Phospho-TSC2-S1387 (rabbit polyclonal) | Abclonal | Cat# AP1117 | WB (1:1000) |
| Antibody | anti-Phospho- TSC2-S939 (rabbit polyclonal) | Cell Signaling Technology | Cat# 3615 | WB (1:1000) |
| Antibody | anti-DYRK1A | PMID:30137413 | | WB (1:2000) |
| Recombinant DNA reagent | pcDNA3-HA3-TSC1 (plasmid) | Addgene | RRID:Addgene_19911 | |
| Recombinant DNA reagent | pcDNA3 Flag TSC2 (plasmid) | Addgene | RRID:Addgene_14129 | |
| Recombinant DNA reagent | LentiCRISPR v2(plasmid) | Addgene | RRID:Addgene_52961 | |
| Sequence-based reagent | DYRK1A-RT-F | This paper | RT-qPCR primers | AAGCTCAGGTGGCTCATCGG |
| Sequence-based reagent | DYRK1A-RT-R | This paper | RT-qPCR primers | TCTCGCAGTCCATGGCCTG |
| Sequence-based reagent | GAPDH-RT-F | This paper | RT-qPCR primers | ACAACTTTGGTATCGTGGAAGG |
| Sequence-based reagent | GAPDH-RT-R | This paper | RT-qPCR primers | GCCATCACGCCACAGTTTC |
| Sequence-based reagent | Control-sgRNA | This paper | SgRNA target sequences for mouse cells | CGAGGTATTCGGCTCCGCG |
| Sequence-based reagent | Dyrk1a-sgRNA | This paper | SgRNA target sequences for mouse cells | CGCTTTTATCGGTCTCCAG |
| Commercial assay or kit | BCA Protein Quantification Kit | Meilunbio | Cat# MA0082 | |
| Commercial assay or kit | HiScript II 1st Strand cDNA Synthesis Kit | Vazyme | Cat# R212 | |
| Chemical compound, drug | Puromycin | Solarbio | Cat# P8230 | |

*Continued on next page*

*Continued*

| Reagent type (species) or resource | Designation | Source or reference | Identifiers | Additional information |
|---|---|---|---|---|
| Chemical compound, drug | Doxycycline | MCE | Cat# HY-N0565 | |
| Software, algorithm | GraphPad | Prism v.7.00 | RRID:SCR_002798 | |
| Software, algorithm | ImageJ | ImageJ v.1.53 | RRID:SCR_003070 | |
| Other | Anti-Flag Beads | Smart-Lifesciences | Cat# SA042005 | |
| Other | r Protein A/G MagPoly Beads | Smart-Lifesciences | Cat# SM015001 | |

## Expression plasmids and cell culture

pcDNA3 -HA3-TSC1 (Yue Xiong; Addgene #19911), pcDNA3 Flag TSC2 (Brendan Manning: Addgene #14129), LentiCRISPR v2 (Feng Zhang; Addgene #52961) plasmids were procured from Addgene. For expressing cDNA using lentivirus, LentiCRISPR v2 was modified by removing the U6 promoter and tracer RNA, and cDNA were cloned in place of SpCas9, and named LentiExp vector. *RHEB* sequence from pRK7-RHEB (John Blenis; Addgene #15888) was cloned into the LentiExp vector by replacing SpCas9. Expression was confirmed by western blot analysis. *DYRK1A* expression vector, *DYRK1A* shRNA, and control shRNA have been previously described (*Li et al., 2018*). All constructs were confirmed by DNA sequencing. Expression of Flag-tagged proteins was induced with 250 ng/ml Doxycycline for 36–48 h. HEK293, HEK293T, and NIH3T3 cells were cultured in Dulbecco's Modified Eagle Medium (DMEM) supplemented with 10% FBS. All cell lines were obtained from the American type culture collection (ATCC). Initially, frozen stocks were tested for mycoplasma after thawing. Cell lines were maintained at 37 °C under 5% $CO_2$.

## Antibodies

Antibodies against Actin (AC026) and HA (AE008) were from Abclonal, anti-DDDDK from MBL (M185-3L), TSC1 (4906), TSC2 (4308), S6K (2708), pS6K (9205), S6 (2317), pS6 (4856) were from CST. DYRK1A antibody was generated in-house and has been described before (*Li et al., 2018*).

## Cell extracts and immunoblotting

For immunoblotting, total cell extracts were prepared. Briefly, collected cells were washed once with cold PBS and lysed in cold NP-40 lysis buffer (40 mM HEPES, pH 7.4, 120 mM NaCl, 1 mM EDTA, 1% NP-40 [Igepal CA-630], 5% glycerol, 10 mM sodium pyrophosphate, 10 mM glycerol 2-phosphate, 50 mM NaF, 0.5 mM sodium orthovanadate and 1:100 protease inhibitors [Solarbio, P6730]) for 15 min on ice. Lysates were cleared by centrifugation, and protein concentration was measured using a BCA kit. Western blot signals were quantified using ImageJ.

## shRNA mediated knockdown and overexpression of RHEB

Short hairpin RNA (shRNA) targeting *DYRK1A* have been previously described (*Li et al., 2018*). Lentiviral particle preparation and infection were performed as previously described (*Li et al., 2018*). Lentivirus-infected HEK293 and SH-SY5Y cells were selected with 1 µg/mL(HEK293) or 2 µg/mL (SH-SY5Y) Puromycin, respectively for 4 days.

For *RHEB* overexpression rescue of *DYKR1A* shRNA-mediated knockdown HEK293 cells, cells were first infected with lentivirus expressing shRNAs targeting control and *DYRK1A*, and then selected with 1 µg/mL Puromycin for 48 hr. After selection cells were transduced with lentivirus expressing *Flag-RHEB*. These cells were further selected with 2 µg/mL Puromycin for 48 hr before harvesting.

## Serum starvation

shRNA-treated HEK293 cells selected for 4 days with 1 ug/ml puromycin were washed once with PBS and then DMEM without serum added. Serum starvation was performed for 12 hr. For stimulation with FBS (10%), pre-warmed FBS was directly added to the serum-free media for the time periods indicated.

## CRISPR/Cas9 mediated *Dyrk1a* knockdown

NIH3T3 cells were transduced with lentiCRISPR v2 (Addgene plasmid #52961) containing either no sgRNA (control) or sgRNA targeting *Dyrk1a* (*Sanjana et al., 2014*). Transduced cells were selected with 3 µg/mL Puromycin for 4 days before harvesting. sgRNA targeting mouse *Dyrk1a* is provided in .

## Cell size measurements

Cell size was measured and analyzed with a JIMBIO FIL electronic cell counter following the manufacturer's protocol. In short, $2.5 \times 10^5$ cells were seeded in six-well plates. After 24 hr of culture, cells were trypsinized and taken up in full DMEM. Each cell line was measured three times; each measurement comprised three cycles of cell counts with intervals of 1 µm ranging from 4–30 µm. The sum of counts of viable cells in the range of 12–30 µm was plotted and quantified. Three biological replicates were performed per cell type (i.e. HEK293, SH-SY5Y, and NIH3T3 cells [KD/KO and corresponding Control]). Control and corresponding KD/KO were compared with multiple unpaired t-tests. p-values are presented as stars above the corresponding bar graphs; *$p<0.05$. HEK293 *Flag-DYRK1A* cell size was measured as above, except cells were induced with 40 ng/mL Doxycycline for 48 hr.

## Quantitative reverse transcription polymerase chain reaction (q-RT-PCR) assays

$2.5 \times 10^5$ cells were seeded in six-well plates. After 48 hr of 40 ng/mL Doxycycline induction, total RNA from fresh cells were extracted using RNA-easy Isolation Reagent (Vazyme) as per the manufacturer's instructions. After determining the total RNA concentration and quality using a Nanodrop (Thermo Fisher), 1 µg RNA was used in 20 µL reaction mixture to reverse transcribe RNA using HiScript II 1st Strand cDNA Synthesis Kit (Vazyme R212) and stored at –20°C. 2 µL was used to perform qPCR using ChamQ Universal SYBR qPCR Master Mix (Vazyme Q711) with 10 µM forward and reverse gene-specific primers. The cycling consisted of 2 min at 95°C, followed by 40 cycles of 5 s at 95°C and 10 s at 60°C. Following the completion of the final cycle, a melting curve analysis was performed to monitor the purity of the PCR products. Each sample was analyzed in triplicate. Quantification was performed by means of the comparative Ct method ($2^{-\Delta CT}$). The mRNA expression of target genes was normalized to that of GAPDH, and the data are represented as the mean ± SEM of biological replicates.

## Mass spectrometry analysis for identification of phosphorylated peptides of TSC1/TSC2

Protein precipitation and digestion: Proteins were precipitated with TCA (trichloroacetic acid). The protein pellet was dried in Speedvac. The pellet was subsequently dissolved with 8 M urea in 100 mM Tris-Cl (pH 8.5). TCEP (final concentration is 5 mM; Thermo Scientific) and Iodoacetamide (final concentration is 10 mM; Sigma) were added to the solution and incubated at room temperature for 20 and 15 min for reduction and alkylation, respectively. The solution was diluted four times and digested with Trypsin at 1:50 (w/w; Promega).

LC/tandem MS (MS/MS) analysis of peptide: The peptide mixture was loaded on a home-made 15 cm-long pulled-tip analytical column (75 µm i.d.) packed with 3 µm reverse-phase beads (Aqua C18, Phenomenex, Torrance, CA) connected to an Easy-nLC 1000 nano HPLC (Thermo Scientific, San Jose, CA) for mass spectrometry analysis. Data-dependent tandem mass spectrometry (MS/MS) analysis was performed with a Q Exactive Orbitrap mass spectrometer (Thermo Scientific, San Jose, CA). Peptides eluted from the LC system were directly electrosprayed into the mass spectrometer with a distal 1.8-kV spray voltage. One acquisition cycle includes one full-scan MS spectrum (m/z 300–1800) followed by the top 20 MS/MS events, sequentially generated on the first to the twentieth most intense ions selected from the full MS spectrum at a 27% normalized collision energy. MS scan functions and LC solvent gradients were controlled by the Xcalibur data system (Thermo Scientific).

## Data analysis

The acquired MS/MS data were analyzed using previously published pipeline (*McCormack et al., 1997*; *Eng et al., 1994*). In order to accurately estimate peptide probabilities and false discovery rates, we used a decoy database containing the reversed sequences of all the proteins appended to

the target database. Carbamidomethylation (+57.02146) of cysteine was considered as a static modification and lysine phosphorylation (+79.9663) of STY as a variable modification.

## Flag immunoprecipitation

With detergent: $1×10^6$ cells were seeded in 10 cm plates. After 24 hr of culture, cells were co-transfected with 5 µg pcDNA-*Flag-DYRK1A*/deletion and 5 µg *HA-TSC1* using PEI. After 48 hr, collected cells were washed once with cold PBS, and lysed in cold NP-40 lysis buffer (40 mM HEPES, pH 7.4, 120 mM NaCl, 1 mM EDTA, 1% NP-40 [Igepal CA-630], 5% glycerol, 10 mM sodium pyrophosphate, 10 mM glycerol 2-phosphate, 50 mM NaF, 0.5 mM sodium orthovanadate, and 1:100 protease inhibitors [Solarbio, P6730]) for 15 min on ice. Lysates were cleared by centrifugation, and protein concentration was measured using a BCA kit. The supernatant was then incubated with Flag beads (Smart-Lifesciences, SA042005) for ~4 hr at 4°C. Finally, beads were washed with a lysis buffer without NP-40 three times and resuspended in a 1X SDS loading buffer. Samples were heated for 10 min at 95°C and separated by SDS-PAGE.

## Overexpressed *DYRK1A* immunoprecipitation without detergent

Approximately $10^8$ HEK293-*Flag-DYRK1A* (with 250 ng/ml Doxycycline treatment), and parental cells were induced for 48 hr were collected and washed with PBS. Cells were swollen for 15 min in hypotonic buffer (Buffer A: 10 mM HEPES pH 7.9, 1.5 mM MgCl2, 10 mM KCl, 0.5 mM DTT supplemented with freshly prepared protease inhibitors, Solarbio, P6730). Swollen cells were dounced 20 times in a Wheaton Dounce homogenizer till about 90% of cells were lysed (as observed in a microscope). Lysates were then centrifuged at 5000–6000 g for 10 min at 4°C. The supernatant was transferred to a new tube, and 0.11 volume of Buffer B (0.3 M HEPES pH 7.9, 1.4 M KCl, 0.03 M MgCl$_2$) was added; the supernatant was then centrifuged at 12,000 g for 30 min. The supernatant was transferred to a new tube and incubated with Flag-beads (Smart-Lifesciences; SA042005) for ~4 hr at 4°C. Finally, beads were washed three times with wash buffer containing 150 mM NaCl, and 0.2% Triton X-100 and resuspended in a 1×SDS loading buffer, and analyzed by SDS-PAGE.

## Endogenous *DYRK1A* immunoprecipitation without detergent

Approximately $10^8$ HEK293 cells were swollen for 15 min in hypotonic buffer (Buffer A: 10 mM HEPES pH 7.9, 1.5 mM MgCl2, 10 mM KCl, 0.5 mM DTT supplemented with freshly prepared protease inhibitors, Solarbio, P6730). Swollen cells were dounced 20 times in a Wheaton Dounce homogenizer and centrifuged at 16,000 g for 30 min. Supernatant was transferred to a new tube, and 0.11 volume of Buffer B (0.3 M HEPES pH 7.9, 1.4 M KCl, 0.03 M MgCl$_2$) was added and incubated with Protein A/G magnetic beads (Smart-Lifesciences, SM015001) for ~4 hr at 4°C. Finally, beads were washed three times with a wash buffer containing 150 mM NaCl and 0.2% Triton X-100 and resuspended in a 1X SDS loading buffer and analyzed by SDS-PAGE.

## Kinase assay

GST-DYRK1A and GST-DYRK1A K188R were cloned in pGEX-6p1vector, and proteins were expressed in BL21 strain and purified using GST beads. FLAG-TSC2 and HA3-TSC1 were co-expressed in HEK293 cells, and Flag-Affinity purified. After wash, beads were rinsed with 1X kinase assay buffer (25 mM HEPES, pH 7.0, 5 mM MgCl2, 0.5 mM DTT), and kinase assay was performed on beads bound with FLAG-TSC2 and HA-TSC1. Equal amounts of GST-DYRK1A or GST-K188R were added to the Flag beads in the presence/absence of ATP (100 µM), and incubated at 30 °C for 30 min. Reactions were stopped by the addition of 5X SDS loading dye and analyzed by SDS PAGE.

## *Drosophila* culture

Flies were grown in standard fly food at 25 °C. Following genotypes were used in this study: Canton S (Bloomington Stock Center), *mnb[1]* (FBal0012364, [*Tejedor et al., 1995*], gift from Francisco J. Tejedor), UAS-*mnb* (FBtp0114512, [*Shaikh et al., 2016*], gift from Francisco J. Tejedor), UAS-*RHEB* (FBst0009688, Bloomington Stock Center), D42-Gal4 (FBti0002759, Bloomington Stock Center [*Gustafson and Boulianne, 1996*], P[(*45*) w[+mC]=UAS Tor.TED]II FBti0026636, [*Hennig and Neufeld, 2002*] similarities in the TOR and, Bloomington Stock Center), UAS-mCherry (FBti0147460, Bloomington Stock Center).

### *Drosophila* NMJ immunofluorescence staining and quantification

For Immunofluorescence staining of NMJ the third instar larvae were fixed in 4% paraformaldehyde (Himedia Cat#TCL119) for 20 min at room temperature and washed in 1 X PBST (0.2% Triton X-100). Antibodies were used at the following dilutions: Mouse anti-DLG 1:500 (DSHB 4F3, *Shaikh et al., 2016*), Rabbit anti-HRP conjugated with alexa488 (dilution 1:500, Jackson 123-545-021), Goat anti-mouse conjugated with Alexa 555 (Dilution 1:500, Invitrogen A28180). Samples were imaged with a confocal microscope (Leica Stellaris 5 PL APO 40 X/1.30 oil objective) or Olympus FV3000 (UPLFLN 40 X/1.30 oil objective). Images were documented and quantified using ImageJ. We marked the HRP-labeled boutons to quantify NMJ phenotypes and manually counted the number. The number of boutons was then normalized to the muscle area (muscle 6/7), and the normalized data were compared between genotypes.

### Statistical analysis

GraphPad Prism version 7.00 was used for statistical and statistical presentation of quantitation. p-values are presented in the figures above or below the compared conditions. Two-way ANOVA followed by a Sidak's multiple comparisons test was applied to cell size data, p-S6, and p-S6k. Unpaired two-tailed Student's *t*-test was applied to DYRK1A protein expression. The data are represented as the mean ± SEM of three biological replicates.

## Acknowledgements

We thank Yue Xiong, Brendan Manning, Feng Zhang, and John Blenis for plasmids and Francisco J Tejedor for *mnb* fly strains. Liang Hu and Meng Huan Zhang for technical help in cloning. Fly strains obtained from the Bloomington *Drosophila* Stock Center (NIH P40OD018537) were used in this study. Funding: This work was supported in part by Yunnan High-end Foreign Experts program to MM, National Natural Science Foundation of China (31471206) to MM. Major Basic Research Project of Science and Technology of Yunnan (202001BC070001) to LPBR. YZH, MPW and LF are supported by The Stowers Institute for Medical Research. MJ is supported by the Department of Atomic Energy (Project Identification No. RTI 4007), Department of Science and Technology, SERB (CRG/2020/003275), Department of Biotechnology (BT/PR32873/BRB/10/1850/2020), Government of India. MJ is a Ramalingaswami fellow, Department of Biotechnology, Government of India, under project number BT/RLF/Re-entry/06/2016. SNJ is supported by DBT/Wellcome trust India Alliance (grant no IA/I/18/1/503629) and intramural funding of CSIR-Centre for Cellular and Molecular Biology, Hyderabad, India.

## Additional information

### Funding

| Funder | Grant reference number | Author |
|---|---|---|
| National Natural Science Foundation of China | 31471206 | Man Mohan |
| Department of Science and Technology, Ministry of Science and Technology, India | CRG/2020/003275 | Manish Jaiswal |
| Department of Atomic Energy, Government of India | RTI 4007 | Manish Jaiswal |
| Department of Biotechnology, Ministry of Science & Technology, India | BT/PR32873/BRB/10/1850/2020 | Manish Jaiswal |

| Funder | Grant reference number | Author |
| --- | --- | --- |
| Stowers Institute for Medical Research | | Ying Zhang Michael Washburn Laurence A Florens |
| Wellcome Trust/DBT India Alliance | IA/I/18/1/503629 | Sonal Nagarkar-Jaiswal |
| Department of Biotechnology, Ministry of Science & Technology, India | BT/RLF/Re-entry/06/2016 | Manish Jaiswal |
| CSIR-Centre for Cellular and Molecular Biology | | Sonal Nagarkar-Jaiswal |

The funders had no role in study design, data collection and interpretation, or the decision to submit the work for publication.

### Author contributions

Pinhua Wang, Formal analysis, Writing – original draft, Writing – review and editing; Sunayana Sarkar, Formal analysis, Validation, Writing – original draft, Writing – review and editing; Menghuan Zhang, Tingting Xiao, Fenhua Kong, Zhe Zhang, Deepa Balasubramanian, Nandan Jayaram, Sayantan Datta, Ruyu He, Ping Wu, Peng Chao, Ying Zhang, Formal analysis; Michael Washburn, Sonal Nagarkar-Jaiswal, Formal analysis, Supervision; Laurence A Florens, Formal analysis, Supervision, Writing – review and editing; Manish Jaiswal, Formal analysis, Supervision, Validation, Writing – original draft, Writing – review and editing; Man Mohan, Conceptualization, Resources, Data curation, Formal analysis, Supervision, Funding acquisition, Validation, Investigation, Visualization, Methodology, Writing – original draft, Project administration, Writing – review and editing

### Author ORCIDs

Michael Washburn ![iD] https://orcid.org/0000-0001-7568-2585
Laurence A Florens ![iD] https://orcid.org/0000-0002-9310-6650
Sonal Nagarkar-Jaiswal ![iD] https://orcid.org/0000-0002-2369-3714
Manish Jaiswal ![iD] https://orcid.org/0000-0001-8360-8289
Man Mohan ![iD] https://orcid.org/0000-0002-8811-8542

Reviewer #2 (Public Review): https://doi.org/10.7554/eLife.88318.3.sa1
Author response https://doi.org/10.7554/eLife.88318.3.sa2

---

# Additional files

### Supplementary files
- MDAR checklist
- Supplementary file 1. Table S1: SgRNA target sequences for mouse cells.

### Data availability

Mass spectrometry data files for FLAG-DYRK1A affinity purifications and negative controls are available from MassIVE (MSV000085815) and ProteomeXchange (PXD020533). Original mass spectrometry data underlying this manuscript can be accessed from the Stowers Original Data Repository.

The following datasets were generated:

| Author(s) | Year | Dataset title | Dataset URL | Database and Identifier |
|---|---|---|---|---|
| Florens L | 2024 | MudPIT analyses of FLAG-DYRK1A and FLAG-Controls affinity-purified from cytoplasmic extracts of HEK293 cells | https://massive.ucsd.edu/ProteoSAFe/dataset.jsp?accession=MSV000085815 | MassIVE, MSV000085815 |
| Wang P, Sarkar S, Zhang M, Xiao T, Kong F, Zhang Z, Balasubramanian D, Jayaram N, Datta S, He R, Wu P, Chao P, Zhang Y, Washburn MP, Florens L, Nagarkar-Jaiswal S, Mohan M | 2024 | DYRK1A Interacts with the Tuberous Sclerosis Complex and Promotes mTORC1 Activity | http://www.stowers.org/research/publications/libpb-1722 | Stowers Original Data Repository, libpb-1722 |

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
