## [Editor Report · eLife assessment]

This **fundamental** study identifies the kinase DYRK1A as a novel component of the tuberous sclerosis complex (TSC) protein complex, which is central to cellular growth and cell size. The findings presented here have broad implications for how cell size and growth is regulated. The methodology and analysis are **convincing** and support the findings.

---

## [Referee Report · Reviewer #2 (Public Review)]

This study reports a physical interaction between the kinase DYRK1A and the Tuberous Sclerosis Complex (TSC) protein complex (TSC1, TSC2, TBC1D7). Furthermore, this study demonstrates that DYRK1A, upon interaction with the TSC proteins, regulates mTORC1 activity and cell size. Additionally, this study identifies T1462 on TSC2 as a phosphorylation target of DYRK1A. Finally, the authors demonstrate that DYRK1A impacts cell size using human, mouse and *Drosophila* cells.

The interaction described here is highly impactful to the field of mTORC1-regulated cell growth and uncovers a previously unrecognized TSC-associated interacting protein. DYRK1A and its regulation of mTORC1 activation may have an impact for multiple diseases in which mTORC1 is hyperactivated.

---

## [Author Response]

The following is the authors’ response to the original reviews.

**Public Reviews:**

**Reviewer #1 (Public Review):**
In this manuscript, Wang et al. demonstrate that knockdown of DYRK1A results in reduced cell size, which is mediated by mTORC1 activity. They found that DYRK1A interacts with TSC1/TSC2 proteins which leads to the phosphorylation of TSC2 at T1462. Phosphorylation of TSC2 at T1462 inhibits TSC2 activity leading to the activation of mTORC1. The authors complement their findings by demonstrating that overexpression of RHEB (positive regulator of mTORC1) rescues the phenotype of DYRK1A (mnb in flies) mutation in the NMJ.The authors' findings on the regulation of cell size and mTORC1 activity by DYRK1A reflect the previous findings of Levy et al. (PMID: 33840455) that cortical deletion of Dyrk1a in mice causes decreased neuronal size associated with a decreased activity of mTORC1 that can be rescued by the inhibition of Pten or supplementation of IGF1.The authors demonstrate that T1462 phospho-site at TSC2 is phosphorylated in response to the overexpression of WT but not kinase-dead DYRK1A. However, the authors do not provide any evidence that the regulation of mTORC1 is mediated via phosphorylation of this site. In addition, T1462 site is known to be phosphorylated by Akt. There is a possibility that Akt was co-purified with TSC1/TSC2 complex and DYRK1A promotes phosphorylation of TSC2 indirectly via the activation of AKT that can be tested by using AKT depleted cells.

We thank the reviewer for reviewing this manuscript and the critical comments. Various groups have reported the significance of the Phosphorylation of TSC2 T1462, along with four other phosphorylation sites, in regulating mTORC1, and therefore, we did not deal with this in the current manuscript (Manning et al. PMID: 12150915, Inoki et al. PMID: 12172553, Zhang et al. PMID: 19593385). Regarding co-purification of AKT with TSC1/TSC2 - AKT phosphorylates T1462, S939 and S1387 (Manning et al. PMID: 12150915, Inoki et al. PMID: 12172553, Zhang et al. PMID: 19593385). However, in in vitro kinase assay, signal intensities of anti-TSC2 S939 and S1387, with or without ATP, showed no significant difference, suggesting that AKT is not pulled down with TSC1 or TSC2. DYRK1A and Kinase dead DYRK1A were expressed and purified from bacteria. Moreover, multiple studies have purified TSC1 and TSC2 and reported no AKT co-purified (Menon et al. PMID: 24529379, Chong-kopera et al. PMID: 16464865).

RHEB is the most proximal regulator of mTORC1 and can activate mTORC1 even under amino acid starvation. The fact that RHEB overexpression rescues the cell size under *DYRK1A* depletion or *mnb* (*DYRK1A* in *Drosophila*) mutant phenotype does not prove that DYRK1A regulates the cell size via TSC1 as it would rescue any inhibitory effects upstream to mTORC1.

We agree with the reviewer that overexpression of RHEB may rescue any inhibitory effects upstream to mTORC1. In the results and discussion sections (Page number 7, last 3 lines), we mentioned that Rheb overexpression only supports our suggestion that DYRK1A likely works upstream to RHEB. We, however, have performed another experiment to strengthen our hypothesis. We show that increased cell size phenotype due to DYRK1A overexpression can be suppressed by inhibiting the TORC1 pathway, suggesting that mTORC1 is necessary for DYRK1A-mediated cell growth. These results are presented in Supplementary Figure 4. The results of two reciprocals of experiments (Suppression of DRYK1A/Mnb loss of function phenotypes by RHEB overexpression and suppression of rescue of DYRK1A Gain of function phenotypes) along with and regulation of TSC phosphorylation by DYRK1A strongly suggests that DYRK1A positively regulates TSC pathway.

**Reviewer #2 (Public Review):**
This study aims to describe a physical interaction between the kinase DYRK1A and the Tuberous Sclerosis Complex proteins (TSC1, TSC2, TBC1D7). Furthermore, this study aims to demonstrate that DYRK1A, upon interaction with the TSC proteins regulates mTORC1 activity and cell size. Additionally, this study identifies T1462 on TSC2 as a phosphorylation target of DYRK1A. Finally, the authors demonstrate the role of DYRK1A on cell size using human, mouse, and *Drosophila* cells.This study, as it stands, requires further experimentation to support the conclusions on the role of DYRK1A on TSC interaction and subsequently on mTORC1 regulation. Weaknesses include, (1) The lack of an additional assessment of cell growth/size (eg. protein content, proliferation), (2) the limited data on the requirement of DYRK1A for TSC complex stability and function, and (3) the limited perturbations on the mTORC1 pathway upon DYRK1A deletion/overexpression.

We thank the reviewer for reviewing this manuscript and the comments. We have previously analyzed the effect of DYRK1A knockdown in the proliferation of THP cells (human leukemia monocytic cell line) (Li Shanshan et al. PMID: 30137413) and have shown that DYRK1A knockdown negatively affects cell proliferation. Other studies have also shown a role for DYRK1A in cell proliferation, including in foreskin fibroblasts (Chen et al. PMID: 24119401) and HepG2 cells (Frendo-Cumbo et al. PMID: 36248734). mTORC1 regulates several pathways, including protein synthesis, lipid synthesis, nucleotide synthesis, autophagy, and stress responses. We have not done the protein content as this parameter is directly affected by TORC1 activation and may not be a suitable measure for cell growth. A large number of studies involving mTORC1 regulation analyze the levels of S6K and S6 phosphorylation, as these are direct readouts of mTORC1 function (Prentzell et al. PMID: 33497611, Zhang et al. PMID: 17052453, Ben-Sahra et al, PMID: 23429703, Düvel et al. PMID: 20670887, Zhang et al. PMID: 2504303). Therefore, we used these markers to assess the status of the mTORC1 pathway.

(2) ..the limited data on the requirement of DYRK1A for TSC complex stability and function,

We agree with this limitation in our study. We have not seen a significant difference in TSC1 or TSC2 protein levels in DYRK1A knockdown or overexpressing cells, so we did not follow up on this aspect.

..and (3) the limited perturbations on the mTORC1 pathway upon DYRK1A deletion /overexpression.

We have performed an additional experiment where we overexpressed DYRK1A and showed that increased cell size phenotype due to DYRK1A overexpression can be suppressed by inhibiting the TORC1 pathway, suggesting that mTORC1 is necessary for DYRK1A-mediated cell growth. These results are presented in Supplementary Figure 4. The results of two reciprocals of experiments (Suppression of DRYK1A/Mnb loss of function phenotypes by RHEB overexpression and suppression of Rescue of DYRK1A Gain of function phenotypes) along with and regulation of TSC phosphorylation by DYRK1A suggests that DYRK1A positively regulates TSC pathway.

Finally, this study would benefit from identifying under which nutrient conditions DYRK1A interacts with the TS complex to regulate mTORC1. The interaction described here is highly impactful to the field of mTORC1-regulated cell growth and uncovers a previously unrecognized TSC-associated interacting protein. Further characterization of the role that DYRK1A plays in regulating mTORC1 activation and the upstream signals that stimulate this interaction will be extremely important for multiple diseases that exhibit mTORC1 hyper-activation.

We agree that identifying nutrients (or physiological conditions) that affect DYRK1A-mediated TSC regulation will be important to understanding the additional complexity in context-dependent mTORC1 activation/deactivation. This study has not addressed those issues, particularly due to DYRK1A's pleiotropic nature. DYRK1A has many substrates, and both overexpression and loss of DYRK1A lead to multiple phenotypes. Identifying nutrient conditions or growth factors that can regulate the activation of DYRK1A is not yet known and would require an independent investigation.

**Reviewer #3 (Public Review):**
The manuscript describes a combination of in vitro and in vivo results implicating Dyrk1a in the regulation of mTORC. Particular strengths of the data are this combination of cell and whole animal (*Drosophila*) based studies. However, most of the experiments seem to lack a key additional experimental condition that could increase confidence in the authors' conclusions. Overall some tantalizing data is presented. However, there are several issues that should be clarified or otherwise addressed with additional data.

We thank the reviewer for reviewing and commenting on this manuscript.

(1) In Figure 1G, why not test overexpression levels of Dyrk1a via western rather than only looking at the RNA levels?

Induced overexpression of DYRK1A was probed by analyzing mRNA levels, as the concentration of Doxycycline used (0-100 ng/ml) did not produce enough protein that could be detected by anti-flag antibody in a western blot. We have modified the sentence (page 5, paragraph 1).

(2) In Figure 2, while there is clearly TSC1 protein in the Dyrk1a and FLAG-Dyrk1a IPs that supports an interaction between the proteins, it would be good to see the reciprocal IP experiment wherein TSC1 or TSC2 are pulled down and then the blot probed for Dyrk1a.

In the revised manuscript, we have provided evidence that TSC1 and TSC2 can interact with endogenous DYRK1A. We have performed immunoprecipitation of affinity-tagged TSC1 or TSC2 and have probed for the enrichment of DYRK1A (Supplementary Figure S2).

(3) Figures 3 A and D tested the effects of Dyrk1a knockdown using different methods in different cell lines. This is a reasonable approach to ascertain the generalizability of findings. However, each experiment is performed differently. For example, in 3A, the authors found no difference in baseline pS6, so they did a time course of treatment to induce phosphorylation and found differences depending on Dyrk1a expression. In 3D, they only show baseline effects from the CRISPR knockdown. Why not do the time course as well for consistency? Also, why the an inconsistency in approaches wherein one shows baseline effects and the other does not? The authors could also consider the pharmacologic inhibition of Dyrk1a activity as well.

We agree that different methods were used in different cell lines to assess the effect of DYRK1A. Since *DYRK1A* is a pleiotropic gene, its manipulation has diverse effects on different cell lines. Also, not all cell types have similar levels of mTORC activity. Hence, we had to adapt to different strategies in different cell types, which accounted for the inconsistency in the methodology. However, various groups have used these methods to determine the activity of mTORC1 by S6 and S6K phosphorylation by both starvations, followed by the stimulation and direct estimation methods in cycling cells (Prentzell et al. PMID: 33497611, Zhang et al. PMID: 17052453, Ben-Sahra et al, PMID: 23429703, Düvel et al. PMID: 20670887, Zhang et al. PMID: 25043031). ShRNA-mediated knockdown in HEK293 cells does not change S6 or S6K phosphorylation levels in actively growing cells, whereas cycling NIH3T3 cells shows a significant reduction in S6 and S6K phosphorylation. As suggested, we used pharmacological inhibition of DYRK1A and 1uM Harmine to treat the HEK293 cells and perform starvation. However, cells treated and starved start to float and die in large numbers. Thus, we did not follow this experiment further.

(4) In Figure 4, RHEB overexpression increases cell size in both Dyrk1a wt and Dyrk1a shRNA treated cells, although the magnitude of the effect appears reduced in Dyrk1a shRNA cells. However, there is the possibility here that RHEB acts independently of Dyrk1a. Why not also do the experiment of Figure 1 wherein Dyrk1a is overexpressed and then knockdown RHEB in that context? If the hypothesis is supported, then RHEB knockdown should eliminate the cell size effect of Dyrk1a overexpression.

We thank the reviewer for suggesting this experiment. We have overexpressed DYRK1A using the inducible HEK293A-Flag-DYRK1A overexpression system and treated cells with mTOR inhibitors (Rapamycin or Torin1). The results are added to the supplementary figure S4. Our results show that the increased cell size phenotype due to DYRK1A overexpression can be suppressed by inhibiting the TORC1 pathway. This suggests that mTORC1 is necessary for DYRK1A-mediated cell growth. This data further supports the hypothesis that DYRK1A is a positive regulator of the mTORC1 pathway.

(5) The discussion should incorporate relevant findings from other models, such as Arabidopsis. Barrada et al., Development (2019), 146 (3).

We have incorporated the findings from Arabidopsis (Barrada et al., Development (2019), 146 (3) PMID: 30705074) in the last paragraph of the discussion section.

**Recommendations for the authors:**

**Reviewer #1 (Recommendations For The Authors):**
(1) To demonstrate that DYRK1A can phosphorylate T1462 phospho-site at TSC2 in the absence of Akt using genetic and pharmacological approaches (by using pan-Akt small molecule inhibitors).

We have performed in vitro kinase assay using recombinant DYRK1A, and affinity purified TSC1/TSC2 from HEK293 cells. However, we have not been able to perform this experiment by overexpression of DYRK1A in human cells, as (1) strong overexpression of DYRK1A leads to cell cycle exit, as demonstrated by various laboratories (Soppa et al. PMID: 24806449, Hämmerle et al PMID: 21610031, Najas et al. PMID: 26137553, Park et al. PMID: 20696760) and our observations, and (2) T1462 Antibody signal is weak and cannot be seen in cellular extracts. We have attempted this experiment with at least three different batches of T1462 antibody from CST without success.

(2) To demonstrate that endogenous phosho-mutant/mimetic substitution of T1462 phospho-site at TSC2 is sufficient to prevent the regulation of cell size/NMJ phenotype in *Drosophila* by DYRK1A (mnb).

This is an interesting experiment, and we thank the reviewer for this suggestion. However, we are skeptical about interpreting the possible results. Since T1462 substitution will also block the regulation by other kinases, e.g., Akt, and it may constitutively suppress the mTORC1, any interpretation will be confusing.

**Reviewer #2 (Recommendations For The Authors):**
(1) In section 2.1 the authors claim that DYRK1A down-regulation enhances cell growth. An additional assessment of cell growth or size would strengthen this statement. Is total protein content also increased upon DYRK1A overexpression? Does DYRK1A KD also increase cell proliferation? In Figure 1, providing the median or mean size of cells in each condition will help the reader understand the impact of DYRK1A on cell size. In Supplementary Figure 1, the important statistical differences should be highlighted.

We have not claimed that down-regulation of DYRK1A enhances cell growth. We have not tested the protein content in a cell directly. Knockdown of DYRK1A leads to a reduction in cell proliferation, as shown by various groups, including ours (Shanshan Li PMID: 30137413, Luna et al. PMID: 30343272). Cell size is a very dynamic process and is variable within the population. All the studies measuring cell size show the size using assays on a population of cells. We have not been able to figure out a way to display the median or mean cell size that accurately reflects the cell size of the whole population.

(2) In section 2.2 the authors describe the interaction between DYRK1A and the TSC proteins. Do the DYRK1A mutants impact interaction with TSC2 and TBC1D7 or is this specific to TSC1?

We have not tested this possibility.

(3) In section 2.3, more detailed perturbations of the mTORC1 pathway are needed. Is the mTORC1 activation observed sensitive to rapamycin treatment? Since mTORC1 regulates cell size via S6 ribosomal protein and transcription via 4EBP1, phosphorylation of 4EBP1 should also be considered. In Figure 3A, what is the level of DYRK1A down-regulation? It is unclear how many shRNA constructs were used or whether these were pooled constructs or single clones. If one shRNA/sgRNA is used, it would be very helpful to validate some of the key findings of this study with at least one more clone.

Many research studies have measured the activity of various mTORC1 substrates, the most commonly used being the phosphorylation of S6 and S6K. We agree that analyzing 4EBP1 would make the study more comprehensive, but to complete the study with our limited resources and in a limited time, we have not attempted to establish the 4EBP1 phosphorylation status. We have used a previously described and validated DYRK1A shRNA (as mentioned in the methods section).

(4) In section 2.3 is T1462 an activating or inhibiting phosphorylation event? If DYRK1A phosphorylates and activates mTORC1 via RHEB, shouldn't that result in the inhibition of mTORC1?

Multiple laboratories have demonstrated that T1462 phosphorylation leads to a reduced TSC complex activity and, hence, increased mTORC1 activity (Manning et al. PMID: 12150915, Inoki, PMID: 12172553, Zhang PMID: 19593385).

(5) In section 2.4, what is the status of AKT phosphorylation? Would an AKT inhibitor be useful in this scenario?

AKT phosphorylates T1462, S939 and S1360, as demonstrated by others. However, in our in vitro assay kinase assay, the following facts suggest that AKT is not involved in T1462 phosphorylation we observed:

(1) Signal intensities of anti-TSC2 S939 and S1387 with or without ATP, do not show any significant differences, suggesting that AKT is not pulled down with TSC1 or TSC2.

(2) Multiple studies have performed phosphorylation studies of TSC1 and TSC2 and have not reported any co-purification of AKT.

(6) Very minor grammar errors were observed, mostly at the beginning of the manuscript.

We tried our best to fix grammatical errors.